# Comparison of Nutritional and Nutraceutical Properties of Burdock Roots Cultivated in Fengxian and Peixian of China

**DOI:** 10.3390/foods10092095

**Published:** 2021-09-04

**Authors:** Xiaoxiao Zhang, Daniela D. Herrera-Balandrano, Wuyang Huang, Zhi Chai, Trust Beta, Jing Wang, Jin Feng, Ying Li

**Affiliations:** 1School of Food and Biological Engineering, Jiangsu University, Zhenjiang 212013, China; 17865814897@163.com (X.Z.); wuyanghuang@hotmail.com (W.H.); 2Institute of Agro-Product Processing, Jiangsu Academy of Agricultural Sciences, Nanjing 210014, China; danieladh19@outlook.com (D.D.H.-B.); sophia_chai@163.com (Z.C.); fengjinzju@163.com (J.F.); 3Department of Food and Human Nutritional Sciences, University of Manitoba, Winnipeg, MB R3T 2N2, Canada; Trust.Beta@umanitoba.ca; 4College of Chemical Engineering, Nanjing Forestry University, Nanjing 210037, China; wjcrystal_gl@163.com

**Keywords:** burdock root, nutrient composition, amino acids, caffeoylquinic acids, volatile compounds, in vitro antioxidant capacity

## Abstract

This study aimed to analyze and compare the nutritional quality of powders of burdock root from Fengxian (FX) and Peixian (PX) in China. The nutrient composition including carbohydrates, protein, amino acids, vitamin C, carotenoids, as well as total phenols, total flavonoids and phenolic compounds were investigated in addition to in vitro antioxidant capacity. The results showed that the basic nutrients of burdock root powder (BRP) in both locations did not have significant differences (*p* > 0.05), although the in vitro antioxidant capacity of BRP of Fengxian (F-BRP) was greater than that of PX (*p* < 0.05). The burdock root peel powder (BRPP) possessed more phenolics and stronger in vitro antioxidant capacity than the burdock root powder (BRP) and peeled burdock root powder (PBRP) (*p* < 0.05). Moreover, better quality burdock root was obtained from FX. F-BRP was consequently analyzed by ultra-performance liquid chromatography tandem mass spectrometry for its phenolic composition. Seventeen phenolics, mainly caffeoylquinic acids, were detected. In addition, a total of 181 volatile compounds belonging to eight types were detected including alcohols, aldehydes, ketones, alkenes, esters, acids, linear or aromatic hydrocarbons, and others. The diverse compounds found in this study can provide a theoretical basis for the development and utilization of burdock in the food industry.

## 1. Introduction

Burdock (*Arctium lappa* L.) is a biennial herbaceous plant belonging to the genus *Arctium* of the Compositae family. Burdock contains proteins, a variety of amino acids, polysaccharides, phenolics, vitamins, and other substances, which explains why it is considered a nutritional food and a health care product [1]. Although radical scavenging plays an important role in disease prevention by phenolics, this is not their only mechanism of action. The main phenolic compounds of the burdock roots are caffeoylquinic acid derivatives [1,2]. However, chlorogenic and caffeic acids, cynarin, quercitrin, arctiin, quercetin, and luteolin have also been found, along with a diversity of other bioactive compounds [1,3,4].

Burdock is popular in the Southeast Asian market for its unique aroma and excellent nutritional value. It is especially popular in Japan and Taiwan. However, in the past few decades, due to the nutritional value, biological activity, as well as health benefits of burdock, this plant has gradually gained international recognition for culinary use [5]. Burdock root contains the dietary fiber inulin, is rich in antioxidant polyphenols, and has in vitro antioxidant activity and anti-inflammatory effects [6,7]. Some in vivo studies, using rats as test subjects, showed the potential use of burdock extracts as beneficial for the promotion of gastric ulcer healing and weight management [8,9]. Over an 8-week period, Hou et al. studied rats consuming a high-fat diet and receiving daily intragastric aqueous burdock extract injections ranging from 2–6 g/kg of bodyweight [9]. These rats had reduced weight and reduced total blood cholesterol compared to those that did not receive doses of the extract. Da Silva et al. showed that the anti-inflammatory properties of burdock could be used to promote healing for gastric ulcers [8]. Rats with acetic acid-induced gastric ulcers were treated with 10 mg of burdock extract/kg of body weight. After a 7-day period, the gastric wound was much smaller, and inflammation was visibly reduced compared to rats without treatment. Due to these benefits and a wide range of pharmacological effects, burdock root may potentially be valuable for wider use and development within the food industry.

Fengxian and Peixian are the largest and second-largest areas of burdock cultivation, respectively, in Jiangsu Province of China. In Fengxian, known as the “Hometown of Burdock”, the cultivation area has reached 3333 hm^2^ and produces 150,000 t annually. Burdock is both inexpensive and versatile. Home cooks can use the roots in a variety of recipes; they can be pickled, boiled, used in salads, stir fries, or soups. Burdock can also be processed for use in a variety of products including teas, cakes, sauces, wine, beverages, and even washing liquid. Fengxian and Peixian in Xuzhou City, Jiangsu Province, have a warm, temperate semi-humid monsoon climate with four distinct seasons. These counties make up part of the Yellow River floodplain and the sandy loam soil of this region, being deep with good water and air permeability, is suitable for root and vegetable cultivation. The annual average temperature in Fengxian is 15.3 °C, the annual precipitation is about 653.3 mm, and the annual sunshine hours are 3863.3 h. The frost-free period is about 187 days. Meanwhile, the annual average temperature in Peixian is 14.2 °C, the annual average precipitation is 816.4 mm, and the annual sunshine hours are 3775.1 h, while the average annual frost-free period is about 201 days. Fengxian and Peixian are the main exporters of burdock. At present, there are 30 burdock processing and export enterprises in Fengxian county, together these have total annual sales revenue of 2 billion RMB (over 300 million USD) which accounts for more than 50% of China’s burdock processing and export. In Hekou town, found in Peixian county, there are more than 20 different burdock enterprises which have a processing capacity of about 40,000 tons, and an annual processing value of about 300 million RMB (over 46 million USD). This research compared the quality of the representative burdock variety “Liuchuanlixiang” widely planted in Fengxian and Peixian. In order to determine which of the two locations produce higher quality burdock roots, differences in nutritional analysis, in vitro antioxidant capacity, vitamin C, carotenoids, amino acids, and total phenolic and flavonoid content of burdock roots grown in Fengxian and Peixian were investigated. In addition, the detailed composition of elements, phenolic compounds, and volatile compounds were analyzed for the higher quality burdock roots from Fengxian.

## 2. Materials and Methods

### 2.1. Chemical and Reagents

Folin–Ciocalteu reagent was purchased from Sigma-Aldrich (Shanghai, China). Fluorescein sodium, 1,1-diphenyl-2-trinitrophenylhydrazine (DPPH), 2,2′-azino-bis (3-ethylbenzothiazoline-6-sulfonic acid) (ABTS), 2,3,5-triphenyl-2h-tetrazole chloride (TPTZ), 2,2-azobis (2-methylpropylimid) acetate (AAPH), and 6-hydroxy-2,5,7,8-tetramethyl-chroman-2-carboxylic acid (Trolox, also known as water-soluble vitamin E) were purchased from TCI Chemicals (Shanghai, China). Carotenoid Kit was purchased from Soleibao Technology Co., Ltd. (Beijing, China). Gallic acid, vitamin C, and rutin were purchased from Shanghai Macklin Biochemical Co., Ltd. (Shanghai, China). Protein and plant soluble sugar kits were purchased from Jiancheng Bioengineering Research Institute Co., Ltd. (Nanjing, China). The standards of amino acids, including aspartic acid, glutamic acid, asparagine, serine, glutamine, histidine, glycine, threonine, citrulline, arginine, alanine, tyrosine, cysteine, valine, methionine, norvaline, tryptophan, phenylalanine, isoleucine, leucine, hydroxyproline, lysine, sarcosine, and proline were provided by Guangzhou Weiping Technology Service Co., LTD (Guangzhou, China). All solvents and reagents were of analytical grade, including ethanol, sodium carbonate, sodium hydroxide, sodium nitrite, aluminum nitrate, dipotassium hydrogen phosphate, potassium persulfate, sodium acetate, ferric chloride, ferrous sulfate, and gallic acid, which were purchased from Beijing Reagent (Beijing, China).

### 2.2. Preparation of Burdock Root Powder and Extract Samples

The “Liuchuanlixiang” variety of burdock was collected from Fengxian and Peixian of Xuzhou City, Jiangsu Province, China. For each county, at least eight mature plants from the burdock garden were randomly selected to collect the roots (about a total of 10 kg) and a portion of these was peeled. Burdock peel, peeled burdock, and whole burdock were taken and dried in an oven at 60 °C. After drying, samples were ground and passed through a 60-mesh sieve to obtain the burdock root powder (BRP), peeled burdock root powder (PBRP), and burdock root peel powder (BRPP) samples. The burdock root phenolic extracts were obtained according to the method described by Wu, Gu, Prior, and McKay [10], which was slightly modified. The powder sample (2 g) was weighed and homogenized. Methanol (80%) was added for solvent extraction. The ultrasonic extraction was performed in an RH7200DB CNC ultrasonic device (Kunshan Ultrasonic Instrument Co., Ltd., Suzhou, China) at 100 W and 20 °C for 20 min. Afterwards, the supernatant was centrifugated (5000 rpm, 10 min) and taken as a phenolic extract from burdock roots. The extraction step was repeated 3 times with 20 mL, 20 mL, and 10 mL of MeOH (80%), and the extracts were combined. The powder and extract samples were both stored at −20 °C for subsequent analysis.

### 2.3. Chemical Composition Analysis

Moisture content was determined according to the direct drying methodology (GB 5009.3-2010). The total vitamin C content was obtained according to the 2,6-dichloroindophenol titration method (GB5009.86-2016). First, one milliliter of standard vitamin C solution was accurately measured and put into a 50 mL conical flask, and 10 mL metaphosphoric acid solution was added. The solution was well shaken, then titrated with 2,6-dichlorophenol solution until pink, with the color lasting for 15 s. At the same time, another 10 mL metaphosphate solution was used for the blank test, then was titrated. One gram of sample was weighed, and 100 mL metaphosphate solution was added. Afterward, 10 mL of solution was accurately extracted into a 500 mL conical flask and titrated with a calibrated 2,6-dichlorophenol solution until the solution turned pink and did not fade for 15 s. At the same time, a blank test was performed. The total vitamin C content in the samples was calculated according to the following equation.
$$X=\frac{\left(V-V_{0}\right)\times T\times A}{m}\times 100$$
where *X* is the total vitamin C content in the samples, *V* is the volume of 2,6-dichlorophenol solution consumed by titration samples (mL), *V*_0_ is the volume of 2,6-dichlorophenol solution consumed by titration of blank samples (mL) *T*: is titration of 2,6-dichlorophenol solution, *A* represents diluted multiples, and *m* is the weight of the samples (g).

The total phenolic content was determined using the Folin–Ciocalteu method [11]. Briefly, 0.4 mL of sample was oxidized with 2 mL of 0.5 mol/L Folin–Ciocalteu reagent at room temperature for 4 min. Then, the reaction was neutralized by adding 2 mL of 75 g/L saturated sodium carbonate. After 2 h of incubation in the dark, the absorbance was read at 760 nm using a Mapada UV-1600PC spectrophotometer (Meipuda Instrument Co., Ltd., Shanghai, China). The quantification was based on the standard curve of gallic acid. The TPC results were expressed in gallic acid equivalent (GAE), i.e., mg GAE/g dry weight (DW). 

The total flavonoid content (TFC) was measured based on the formation of a flavonoid–aluminum complex [12]. Briefly, 1 mL of sample was mixed with 0.1 mL of 5% NaNO_2_ for 6 min. Then, 0.1 mL of 10% AlCl_3_·6H_2_O solution was added to the mixture for another 5 min. After adding 1 mL of 1 mol/L NaOH, the reaction solution was mixed well. The absorbance was measured at 510 nm. Rutin was used as a standard in order to establish the calibration curve. The TFC was calculated and expressed in rutin equivalent (RTE), i.e., mg RTE/g DW. 

Quantities of carotenoids, proteins, and soluble sugars (representing carbohydrates) were all determined with their respective kits. Burdock root powder from Fengxian (F-BRP) was selected for evaluation of the content of elements using inductively coupled plasma mass spectrometry (ICP-MS) (GB 5009.268-2016). A sample of ~0.2–0.5 g was added to the microwave digestion tank, ~5–10 mL nitric acid was added, and the tank was covered for 1 h or overnight. The digestion was carried out according to the standard operating procedures of microwave digestion instrument.

The digested sample was cooled down before opening the tank cover to exhaust gases. The inner cover was rinsed with a small amount of water and the digestion tank was placed in an ultrasonic bath, degassed with ultrasounds for 2–5 min. The volume was made up to 50 mL with water and the sample was mixed. A blank test was performed at the same time using this procedure. The mixed standard solution was injected into the inductively coupled plasma mass spectrometer to determine the signal response values of the elements to be measured and the internal standard elements. The concentration of the elements to be measured was the x-coordinate, and the ratio of the response signal values of the elements to be measured and the selected internal standard elements was the y-coordinate. The blank solution and the sample solution were injected into the ICP-MS, and the signal response values of the elements to be measured and the internal standard elements were determined. The concentration of the elements to be measured in the digestion solution was obtained according to the standard curve.

### 2.4. Determination of Amino Acids

An acid hydrolysis methodology previously reported by Kriukova et al. [13] with slight modification was used for the determination of amino acids. A sample of 0.2 g BRP was weighed and put into a sealed bottle, and 10 mL of 6 M HCl containing 1% phenol were added. Nitrogen was then added for one minute and the bottle was sealed prior to hydrolysis at 110 °C for 22 h. After the time had elapsed, the contents were taken out, cooled and diluted to 10 mL. First, one milliliter was taken and dried with 95 °C nitrogen gas, and then one more milliliter of 0.01 M HCl was added for dissolution and the solution was permeated through a membrane for measurement. The amino acid determination was performed using an Agilent automated on-line derivatization method. The primary amino acids were derivatized with o-phthalaldehyde (OPA), while the secondary ones were derivatized with fluorene methoxycarbonyl chloride (FMOC) before being passed through the column for detection. HPLC analysis was conducted in an Agilent-1100 (Agilent Technologies, Santa Clara, CA, USA) equipped with a diode array detector (DAD). The separation was carried out on a ZORBAX Eclipse AAA (4.6 × 75 mm, 3.5 μm) column, with a flow rate of 1.0 mL/min. Mobile phase A was 40 mM NaH_2_PO_4_ (pH 7.8), and mobile phase B was acetonitrile/methanol/H_2_O (45:45:10, *v*/*v*/*v*). The elution gradient was as follows: 0% B (from 0 to 1 min), 0% to 57% B (1 to 23 min), 57% to 100% B (23 to 27 min), 100% B (27 to 34 min), 100–0% B (34 to 40 min), 0%B (40 to 41 min). The detection was conducted with UV at 338 nm and with fluorescence using 266 nm excitation and 305 nm emission. Fluorescence detection was used for proline, hydroxyproline, and sarcosine, and UV detection was used for other amino acids (338 nm). 

### 2.5. In Vitro Antioxidant Capacity

The in vitro antioxidant capacity of burdock root extracts was evaluated by DPPH, ABTS, and FRAP assays. The scavenging activity for DPPH radical was determined by following the method by Wu et al. [14]. The measurement of ABTS cation radical scavenging ability was performed using the methodology reported by Cai et al. [15]. The oxygen radical absorbance capacity (ORAC) assay was conducted according to the method by Dávalos et al. [16]. The assays mentioned above were expressed as Trolox equivalent antioxidant capacity (TEAC)/g DW. The ferric reducing antioxidant capacity power (FRAP) was carried out following the method by Dang et al. [17]. The FRAP results were expressed as Fe (II) equivalent antioxidant capacity (FEAC)/g DW.

### 2.6. Ultra Performance Liquid Chromatography Tandem Mass Spectrometry (UPLC-MS/MS) for Identification of Phenolic Composition

The extracts were subjected to a Waters Acquity UPLC (I-class)-MS/MS system (Waters Corporation, Massachusetts, USA) equipped with a DAD and a Waters Xevo TQ-S Micro using a column Waters ACQUITY UPLC BEH C18 (2.1 × 50 mm) at 35 °C. Mobile phase A was H_2_O (containing 0.25% formic acid), and mobile phase B was acetonitrile (0.25% formic acid). The elution gradient was as follows: 5% B (from 0 to 1 min), 5% to 25% B (1 to 5 min), 25% to 60% B (5 to 9 min), 60% to 100% B (9 to 16 min), and 100% to 5% B (16.2 to 19 min). The injection volume was 1 μL and a flow rate of 0.3 mL/min was set. The phenolic compounds were detected with DAD at 280 nm. ESI-MS analysis was calibrated for positive ion mode (ESI+) and negative ion mode (ESI−) in a mass range of 100–1000 (*m*/*z*) under the following conditions: a spray voltage of 2.0 kV, a cone voltage of 30 V, a solvent removal temperature of 500 °C, and a solvent removal gas (N_2_) flowing at 1000 L/h. Phenolic compounds were tentatively identified by comparing the retention time, mass ion pattern, and UV information with the data from previously reported literatures [2,18].

### 2.7. Determination of Volatile Compounds

The method described by Chai et al. [19] was used for the separation and identification of volatile compounds of F-BRP sample. A 5 g of sample was placed in an extraction bottle and was put in a water bath at 60 °C for 30 min with magnetic stirring (300 rpm). The extraction needle was activated at the gas injection inlet for 20 min (250 °C) before use. An Agilent 7890A-5975C GC-MS system (Agilent Technologies, CA, USA) was used, equipped with a DB-WAX capillary column (50 m × 0.25 mm × 0.25 µm). Helium was used as the carrier gas, with a flow rate of 1.5 mL/min. The oven temperature was programmed from 50 °C and was gradually increased and adjusted to different exposure times. We set the initial temperature at 50 °C for 1 min and raised the temperature at a rate of 5 °C/min to 100 °C and held at this temperature for 5 min. Afterwards, the temperature was increased at a rate of 4 °C/min to 140 °C and held for 10 min, further to 180 °C for 10 min. Finally, the temperature was increased at the same rate to 250 °C and this temperature was maintained for 5 min. Samples were injected into the GC at 250 °C. MS was operated in full scan mode (mass range, *m*/*z* 50–550) with ionization voltage of 70 eV, and an ion source temperature of 230 °C. A library search was performed using the in-house database of NIST11.L.

### 2.8. Statistical Analysis

All experiments were performed in triplicate, and the results were expressed as mean ± standard deviation (SD). One-way analysis of variance (ANOVA) with Tukey’s test was used to determine statistical differences among burdock root samples. Differences were considered significant at *p* < 0.05.

## 3. Results and Discussion

### 3.1. Nutritional Analysis of Burdock Root Powder Samples

The burdock root samples used in the study were collected from Fengxian and Peixian. As shown above, Fengxian has a greater number of light hours, higher average temperature, and lower annual precipitation than Peixian. These differences in geographical conditions between the two locations may lead to differences in basic nutrients and non-nutrients. The moisture content of burdock root in Fengxian accounted for 77.68 ± 0.70% of fresh weight (FW), while it accounted for 77.69 ± 0.82% in Peixian (Figure 1). Hence, statistically significant differences were not found regarding the moisture content of BRP from both locations (*p* > 0.05). Other authors have found that light and annual precipitation affect the sugar content [20,21], which was consistent with our findings, where the carbohydrate content of burdock root powder in Fengxian (542.84 ± 44.19 mg/g) was higher (*p* < 0.05) than those found in Peixian (484.61 ± 4.13 mg/g) (Figure 1). Protein and amino acid content were positively correlated with accumulated temperature and sunshine hours [22]. The content of proteins was slightly higher in F-BRP with 68.47 ± 14.25 mg/g compared to 65.68 ± 12.96 mg/g in P-BRP (Figure 1, *p* > 0.05). Burdock roots are a low-fat food. The lipid contents in BRP from Fengxian and Peixian were only 10.70 ± 1.08 mg/g and 10.33 ± 0.44 mg/g, respectively (Figure 1). The basic nutrients in burdock roots from Fengxian and Peixian had no pronounced differences except for carbohydrates, which may be attributed to their non-significant environmental discrepancy. Song et al. previously reported that the total amount of protein plus oil and most of the amino acids were positively correlated with an accumulated temperature and the mean daily temperature but were negatively correlated with hours of sunshine and diurnal temperature range [22]. The detailed amino acids in plants might vary significantly depending on the location; therefore, amino acids in BRP samples were further analyzed. 

### 3.2. Determination of Amino Acids in Burdock Root Powder Samples

The nutritional value of protein in food mainly depends on the types of essential amino acids, the content of essential amino acids, and the composition ratio of each essential amino acid [22]. The amino acid composition analysis of BRP from Fengxian and Peixian are shown in Table 1. Tryptophan is destroyed under acidic conditions and therefore was not detected. Moreover, citrulline, hydroxyproline, norvaline, sarcosine, asparagine, and glutamine are unstable under acidic conditions. Therefore, only 16 amino acids were detected in this study. These results showed similar total amino acid content of 80.763 mg/g and 77.762 mg/g determined in F-BRP and P-BRP, respectively. F-BRP had arginine in highest quantities, followed by glutamic acid and aspartic acid, whereas P-BRP burdock had glutamic acid in highest quantities, followed by arginine and aspartic acid. Arginine is an amino acid essential for fetal and neonatal nutrition, and it is very important for the detoxification and synthesis of ammonia (including creatine, nitric oxide, and polyamines) [23]. Glutamic acid is a functional amino acid which plays a key role in protein structure, nutrition, metabolism, and signal transduction [24]. The synthesis of aspartic acid, which drives the increase in biomass, maintains a high level of cardiac function, prevents myocardial hypertrophy, and improves myocardial energy characteristics [25]. The whole burdock powder contains seven pharmacodynamic amino acids, including arginine, glutamic acid, aspartic acid, phenylalanine, leucine, isoleucine, and lysine, accounted for more than 77% of the total amino acids, indicating that burdock has high nutritional and medicinal value. Amino acids with a perceived umami (Glu and Asp) or bitter (Val, Ile, Leu, Phe, His, and Arg) taste accounted for more than 35% of the total amino acid content whereas sweet-tasting amino acids (Ser, Gly, Ala, and Pro) accounted for more than 20%. These amino acids mentioned above give burdock a unique flavor. Six essential amino acids were detected in burdock root powder. The amino acid content of F-BRP (20.65%) was slightly higher than that of P-BRP (19.99%). Regarding the environmental conditions of both locations, the latitude of Fengxian is slightly higher than that of Peixian. As the latitude increases, the total amino acids and essential amino acids show a decreasing trend, but the proportion of essential amino acids to total amino acids shows an increasing trend, indicating that there are factors conducive to the formation and accumulation of essential amino acids in burdock roots.

### 3.3. Chemical Composition and In Vitro Antioxidant Activity of BRP Samples

Being the main contributors to in vitro antioxidant capacity, the total phenolic content and flavonoids are important indicators of the total in vitro antioxidant capacity [14]. Table 2 shows the total phenolic content, total flavonoids and in vitro antioxidant activity (DPPH, ABTS, FRAP, and ORAC) of different types of burdock powder from Fengxian and Peixian. Carotenoids are the precursors of vitamin A and play an important role in anti-oxidation, immune regulation, and anti-cancer properties of foods [26]. Vitamin C is an essential nutrient; in the human body, it is a highly efficient antioxidant that protects it from endogenous and exogenous oxidation pressure [27]. Vitamin C is an important cofactor of gene regulatory enzymes and it participates in many important biosynthetic processes in addition to its key role in immune regulation [27,28]. The vitamin C content of BRP and peeled burdock root powder (PBRP) was equivalent, while in burdock root peel powder (BRPP), it was about 7 times higher than other burdock powder samples, indicating that BRPP is also rich in active substances. On the other hand, the carotenoid content of BRP was higher than peeled burdock root powder and was about 5 times higher in BRPP.

The total phenolic content of F-BRP was 1.23 times higher than that of P-BRP, and its total flavonoid content is roughly the same as that of PX. The content of phenolics and flavonoids in in PBRP was more than 4 times higher than in BRP indicating that phenols are mainly in the burdock root peel. There was significantly higher phenolic content in burdock from Fengxian than from Peixian (BRP, PBRP, and BRPP, all *p* < 0.05). In terms of flavonoid content, there were no statistically significant differences (*p* > 0.05) between Fengxian and Peixian, but there was a slight difference between the two locations when evaluating PBRP and BRPP. The burdock root peel powder from FX had 1.9 mg/g more flavonoids than found in Peixian. Woznicki et al. [29] reported that temperature and light intensity are the main environmental factors that affect the concentration of phenolic substances in plants growing in natural environments. The annual average temperature and sunshine hours of Fengxian are slightly higher than those of Peixian, and the annual precipitation is lower than that of Peixian. Hence, it could be possible that within a certain range, longer sunshine hours, higher temperatures, and less precipitation led to better synthesis and accumulation of phenolic compounds. Therefore, the burdock root in Fengxian had higher total phenolic content.

Different BRP samples showed different in vitro antioxidant capacity (Table 2). Burdock root peel powder had the greatest (*p* < 0.05) free radical scavenging ability in all the performed assays. In general, the DPPH, FRAP, ABTS, and ORAC results of different powder samples showed a trend where burdock root peel > burdock root > peeled burdock root. The in vitro antioxidant capacity of F-BRP was higher than that of P-BRP. Regarding the DPPH results, BRPP from Fengxian and Peixian had 109.76 ± 10.14 and 94.71 ± 1.42 mg TEAC/g DW, respectively, which is about 2.5 times higher than BRP and 3 times that of PBRP. For the FRAP assay, the value of F-BRP was 4.52 ± 0.21 mg FEAC/g DW, which is greater (*p* < 0.05) than P-BRP. Both peeled burdock root and burdock root peel from Fengxian also showed higher TEAC than those of Peixian. ORAC results of burdock root powders were also greater in FC. ORAC values of BRPP indicated that Fengxian had greater (*p* < 0.05) in vitro antioxidant capacity than Peixian samples, with 144.38 ± 5.76 and 94.63 ± 2.53 mg TEAC/g DW, respectively. Moreover, when in vitro antioxidant capacity was evaluated by ABTS assay, BRPP was greater than BRP and PBRP; however, the latter two had similar values. Unlike other in vitro antioxidant capacity assays, ABTS showed that BRPP from Fengxian had lower (*p* < 0.05) in vitro antioxidant capacity than Peixian. Nevertheless, BRP and PBRP from Fengxian still had greater in vitro antioxidant capacity than those from Peixian.

Evaluating in vitro antioxidant capacity by different methodologies generally impacts the final results. Our observation was that the DPPH and ABTS free radical scavenging abilities of burdock root powders in both locations were relatively small, while ORAC and FRAP showed greater variation due to different antioxidant mechanisms. Therefore, the scavenging ability of a sample against one free radical does not represent scavenging ability against other free radicals or the total in vitro antioxidant activity. The ABTS method not only quickly and reliably determines the total in vitro antioxidant capacity in vitro, but also has good applicability to hydrophilic and lipophilic antioxidant reagents or systems [15]. The ORAC test is a reliable test based on inhibiting the oxidation induced by peroxyl free radicals, which is triggered by the thermal decomposition of azo compounds (such as AAPH) [30]. As a stable lipophilic free radical, the DPPH radical is commonly employed in order to evaluate the free radical scavenging potential of plant extracts [11]. FRAP is evaluated by the Fe^3+^ to Fe^2+^ transformations and acts as an important indicator of in vitro antioxidant activity [31]. According to the four assays performed, F-BRP had higher in vitro antioxidant capacity than P-BRP. According to this research, the in vitro antioxidant capacity of burdock root is correlated with total phenols and flavonoids, which is consistent with other studies. Lou et al. [32] also found that the in vitro antioxidant capacity of burdock leaf extract was significantly and positively correlated with the content of total phenols. Based on this statement, the main factors affecting the in vitro antioxidant capacity of burdock root could be the total phenolic content and flavonoids. The carotenoid content in BRPP was significantly lower than that in PBR and PBRP, which was the opposite of other indicators. The vitamin C, total phenols, and total flavonoids of BRPP were significantly higher than those of PBRP and BRP. The results showed that the content of carotenoids had no correlation with in vitro antioxidant activity, indicating that the carotenoid compounds might not be major contributor to antioxidant activity. A weak correlation was also observed between carotenoids content and antioxidant capacity in the extracts from *Citrus* cultivars and Moroccan marine microalgae [33,34].

Upon completion of the above comparative analysis of the nutritional components and in vitro antioxidant capacity of BRP samples from Fengxian and Peixian, it was found that F-BRP had higher quality than P-BRP. Therefore, in order to develop antioxidant and nutraceutical products, burdock root from Fengxian would serve as raw material with great potential. A more in-depth analysis of the nutritional components and phenolic compounds of burdock root in Fengxian was further investigated.

### 3.4. Phenolic Composition of F-BRP

A preliminary identification of phenolic compounds in F-BRP was performed by UPLC-ESI-MS. Figure 2 shows UPLC chromatograms (280 nm) of phenolic extracts from burdock root powder. Furthermore, the dominant peaks in UPLC profile were identified by ESI-MS and by comparison with the related references (Table 3). A total of 17 main peaks were found at 280 nm wavelengths. According to the molecular weight analysis presented in Table 3, phenolic compounds in burdock root powder were mainly caffeoylquinic acids and their derivatives, including two monomers, 3-caffeoylquinic acid (chlorogenic acid) and 4-caffeoylquinic acid (cryptochlorogenic acid); three dimers, 3,4-di-*O*-caffeoylquinic acid, 1,5-di-*O*-caffeoylquinic acid, and 3,5-di-*O*-caffeoylquinic acid; two maloyl derivatives, 1,5-di-*O*-caffeoyl-4-*O*-maloylquinic acid and 3,4-di-*O*-caffeoyl-5-*O*-maloylquinic acid; and two succinoyl derivatives, 1,5-di-*O*-caffeoyl-3-*O*-succinoylquinic acid and 1,5-di-*O*-caffeoyl-4-*O*-succinoylquinic acid. Caffeoylquinic acids made up the majority of the phenolic compounds found in the burdock genus reported by other studies [2,18]. The findings are in agreement with other authors [5,35,36,37]. Phenolic compounds have considerable antioxidant properties [5]. Chlorogenic acid and caffeic acid are mainly present in the burdock root peel, and the content of chlorogenic acid was much higher than that of caffeic acid, which is in accordance with previous reports [5,6]. As Figure 2 shows, peak 4 was chlorogenic acid and peak 17 was identified as caffeic acid. Chlorogenic acid and other caffeoylquinic acid derivatives in burdock roots are major contributors to the antioxidant capacity of the BRP extracts. 

### 3.5. Element Content

Burdock root contains a variety of elements (Table 4), of which potassium (K) is the macro element with the highest content, followed by calcium (Ca), phosphorus (P), magnesium (Mg), and sodium (Na) (10,897.38 µg/g, 4205.58 µg/g, 3176.83 µg/g, 2424.32 µg/g, and 751.57 µg/g, respectively). Potassium and sodium are essential for regulating osmotic pressure in cells and the acid–base balance of body fluids to maintain myocardial function. Calcium is an essential macro element for the human body as it plays an important role in promoting the development of human bones and maintaining the normal function of the heart [38]. Magnesium is an element necessary for the structure and function of bone cells and participates in a variety of physiological metabolic processes [39]. Other elements found in lower quantities include manganese (Mn), zinc (Zn), iron (Fe), and copper (Cu). Manganese is an important life element, which constitutes several enzymes with important physiological functions in the body [40]. As an essential micronutrient, chromium (Cr) plays an important role in all insulin regulation activities and it is an important blood sugar regulator [41]. However, Cr (VI) (chromate, chromium in its hexavalent state) is genotoxic and carcinogenic, which could induce inhalation and lung cancer [42]. Therefore, there is a balance in the amount of supplemental Cr and its oxidation state should be avoided. Zinc is an essential nutrient for human health and has antioxidant, anti-stress, and anti-inflammatory effects. Moreover, zinc is also related to the activity of a variety of enzymes in the human body and maintains the normal physiological functions of the body [43]. Iron is the main component of hemoglobin, with a wide range of physiological functions including maintenance of normal hematopoietic and immune functions [43]. Copper is an important factor involved in immune functioning and biochemical metabolism, which is an active factor in hemoglobin synthesis, having anti-inflammatory and antibacterial effects. He et al. explored the different quantities of calcium, magnesium, zinc, and iron in the roots of different varieties of burdock and found that the calcium content of variety “Liuchuanlixiang” was the highest [44]. Cadmium (Cd) and lead (Pb) are considered harmful elements, while beryllium (Be), nickel (Ni), cobalt (Co), and antimony (Sb) are carcinogens. The contents of these harmful and carcinogenic elements were lower than the national standard, indicating that burdock is a safe food for human consumption. According to our findings, burdock root powder contains a large amount of various macro elements and a small amount of trace elements necessary for the human body, and can be used as a good mineral supplement.

### 3.6. Volatile Composition of F-BRP

Table 5 and Figure 3 show the volatile compounds found in burdock root powder from Fengxian. A total of 181 volatile components belonging to eight categories were detected in F-BRP, and their proportions decreased in the order: linear/aromatic hydrocarbons > alkenes > aldehydes > others > alcohols > acids > esters > ketones. The linear/aromatic hydrocarbons and alkenes accounted for more than 48% of the volatile compounds, indicating that hydrocarbons are the main aroma components of BRP. The highest in proportion among linear or aromatic hydrocarbons was tetradecane, which accounted for 2.98% of the total. Regarding alkenes, the greatest contributor was (Z,Z,Z)-1,8,11,14-heptadectetraene, which made up 11.35% of the total. Phenethyl alcohol, a fruity and floral smelling compound, accounted for 1.00% of the total [45]. Alcohols and phenols usually have woody, musky, and floral aromas [46]. The highest in proportion among aldehyde compounds was hexanal comprising 4.64% of the total. Benzyl alcohol and benzaldehyde are both produced from cinnamic acid under the catalysis of phenylalanine and related biological enzymes. Benzyl alcohol has a sweet and fruity taste, while benzaldehyde has a unique almond taste [47]. As for the acid compounds, hexanoic acid had the highest levels, accounting for 3.05% of the total. Hexanoic acid is an important aroma component in Chinese flavor liquor [48]. The highest in proportion among ester compounds was 2,2,4-trimethyl-1,3-pentanediol diisobutyrate (1.05%). The highest in proportion among ketone compounds was (E,E)-3,5-octadiene-2-one (2.28%). An antioxidant, 4-methyl-2,6-di-tert-butyl, was found in a relatively high proportion (2.19%), and is widely used in food and food-related products. Hydrocarbons were the most abundant aroma components in BRP, followed by aldehydes. They are important flavor substances in burdock. Although the quantities of alcohols, acids, esters, ketones, and other substances were low, they each play a role in the formation of the particular burdock flavor.

## 4. Conclusions

In this study, burdock roots from Fengxian and Peixian were analyzed showing their nutritional quality differences. In general, the nutrient and phytochemical content of F-BRP was slightly higher than P-BRP, probably due to Fengxian’s temperature and precipitation. The weather conditions may be more suitable for the accumulation of phenolic compounds in burdock root, which are positively correlated with in vitro antioxidant capacity. Besides, the latitude of FC was slightly higher than that of PC, leading to speculations that high latitude is beneficial to the formation and accumulation of essential amino acids in BRP. Other environmental conditions, such as the type of soil, pH value and, even the microbes might influence the nutrient and phytochemical content, whose effects will be further analyzed in the future. Regarding chemical composition and in vitro antioxidant capacity, the samples followed the trend BRPP > BRP > PBRP. After an in-depth exploration of the nutritional composition characteristics of F-BRP, elements such as K, Ca, P, Mg, Na, Fe, Zn, and others were found in burdock roots, which are necessary for the human body. A total of 181 volatile components in BRP were detected, which belonged to eight categories. By selecting burdock roots from two different locations for nutraceutical food production, our study suggests that burdock roots from Fengxian could have great potential in the industry. The burdock root in Fengxian may be more suitable for direct consumption. However, our findings do not deny the industrial production value of burdock in Peixian. More regional samples should be selected for further analysis in the future.

## Figures and Tables

**Figure 1 foods-10-02095-f001:**
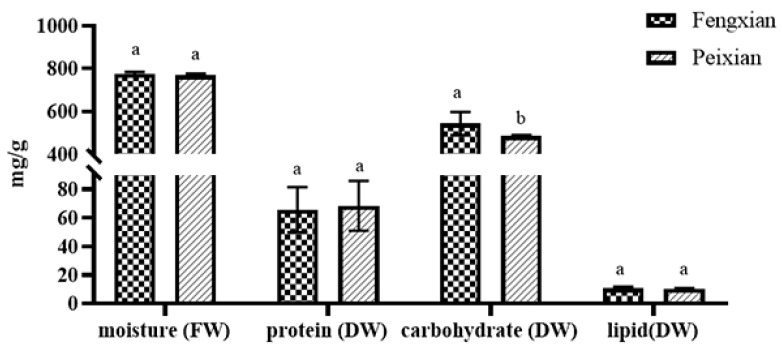
Moisture, protein, carbohydrate, and lipid contents of burdock root samples in Fengxian and Peixian. Different letters in the same column indicate significant differences (*p* < 0.05).

**Figure 2 foods-10-02095-f002:**
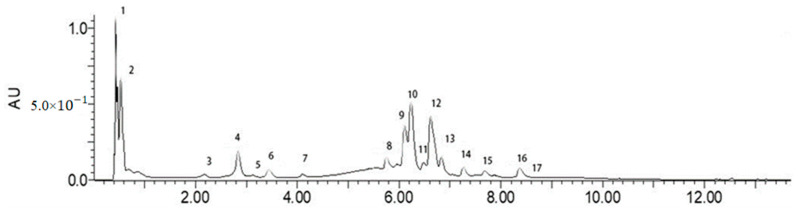
UPLC chromatograms (280 nm) of phenolic extracts from burdock root powder.

**Figure 3 foods-10-02095-f003:**
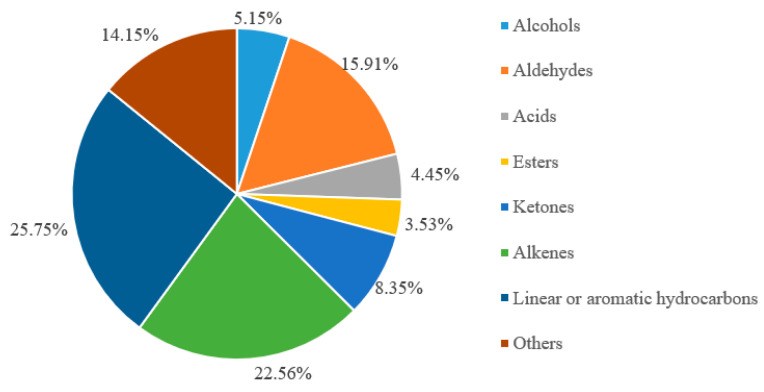
Proportion of volatile compounds in burdock root powder.

**Table 1 foods-10-02095-t001:** Amino acid content of burdock root samples in Fengxian and Peixian.

Amino Acid ^1^	F-BRP	P-BRP
Aspartic acid (Asp)	11.062	13.014
Glutamic acid (Glu)	18.289	17.656
Serine (Ser)	3.382	3.165
Histidine (His)	0.806	0.848
Glycine (Gly)	2.659	2.366
Arginine (Arg)	19.696	16.526
Alanine (Ala)	1.978	1.971
Tyrosine (Tyr)	2.151	1.735
Cysteine (Cys)	1.035	1.459
Proline (Pro)	6.420	5.650
Valine (Val)	2.279	2.368
Phenylalanine (Phe)	4.371	4.512
Isoleucine (Ile)	2.000	2.148
Leucine (Leu)	2.784	2.884
Lysine (Lys)	1.849	1.460
Threonine (Thr)	2.868	2.690
EAA	16.151	16.062
NAA	67.478	64.390
PAA	62.710	60.566
SAA	14.439	13.152
UAA	29.351	30.670
BAA	31.936	29.286
TAA	80.761	77.762

^1^ Content of amino acids (mg/g DW); F-BRP: Fengxian burdock root powder; P-BRP: Peixian burdock root powder; EAA: essential amino acids, were calculated as the total content of Val, Phe, Ile, Leu, Lys, and Thr; NAA: nonessential amino acids, were calculated as the total content of Asp, Glu, Ser, His, Gly, Arg, Ala, Tyr, Cys, and Pro; PAA: pharmacodynamic amino acids, containing Asp, Glu, Ile, Leu, Phe, Lys, and Arg; SAA: sweet amino acids, containing Ser, Gly, Ala, and Pro; UAA: umami flavor amino acid, containing Glu and Asp; BAA: bitter amino acids, containing Val, Ile, Leu, Phe, His, and Arg; TAA: total amino acids.

**Table 2 foods-10-02095-t002:** Chemical composition and in vitro antioxidant capacity of burdock root samples from Fengxian and Peixian.

Sample ^1^	Carotenoid	Vitamin C	TPC	TFC	DPPH	ABTS	FRAP	ORAC
	(mg/g DW)	(mg/g DW)	(mg GAE/g DW)	(mg RTE/g DW)	(mg TEAC/g DW)	(mg TEAC/g DW)	(mg FEAC/g DW)	(mg TEAC/g DW)
Fengxian								
BRP	10.08 ± 0.66 ^a^	1.76 ± 0.18 ^c^	186.20 ± 0.76 ^c^	8.98 ± 0.10 ^b^	40.20 ± 0.66 ^c^	10.88 ± 0.21 ^c^	4.52 ± 0.21 ^c^	66.78 ± 1.96 ^c^
PBRP	6.03 ± 0.18 ^b^	1.77 ± 0.43 ^c^	156.33 ± 1.04 ^d^	6.23 ± 0.11 ^c^	29.29 ± 0.24 ^d^	8.21 ± 0.49 ^d^	3.21 ± 0.12 ^d^	28.46 ± 1.19 ^e^
BRPP	1.95 ± 0.23 ^d^	13.02 ± 1.61 ^a^	776.46 ± 2.86 ^a^	43.65 ± 2.68 ^a^	109.76 ± 10.14 ^a^	41.24 ± 0.75 ^b^	26.51 ± 0.24 ^a^	144.38 ± 5.76 ^a^
Peixian								
BRP	10.24 ± 0.65 ^a^	1.65 ± 0.18 ^c^	151.31 ± 1.04 ^e^	8.62 ± 0.07 ^b^	38.76 ± 1.42 ^c^	8.70 ± 0.18 ^d^	2.89 ± 0.12 ^d^	47.76 ± 1.22 ^d^
PBRP	4.89 ± 0.45 ^c^	1.49 ± 0.16 ^c^	127.53 ± 0.58 ^f^	6.60 ± 0.37 ^c^	29.69 ± 1.12 ^d^	7.26 ± 0.33 ^e^	2.09 ± 0.09 ^e^	19.83 ± 2.40 ^f^
BRPP	2.04 ± 0.31 ^d^	12.31 ± 1.53 ^b^	722.14 ± 3.64 ^b^	41.75 ± 1.53 ^a^	94.71 ± 1.42 ^b^	49.86 ± 0.32 ^a^	22.87 ± 0.76 ^b^	94.63 ± 2.53 ^b^

^1^ BRP, PBRP, and BRPP indicate the burdock root powder, peeled burdock root powder, and burdock root peel powder, respectively. TPC and TFC are the abbreviations of total phenolic content and total flavonoid content, respectively. DPPH, ABTS, FRAP, and ORAC are the abbreviations of the DPPH radical scavenging activity, ABTS radical cation scavenging activity, ferric reducing antioxidant power, and oxygen radical absorbance capacity, respectively. GAE, RTE, TEAC, and FEAC represent the gallic acid equivalent, rutin equivalent, Trolox equivalent antioxidant capacity, and Fe (II) equivalent antioxidant capacity, respectively. Different letters (^a^–^f^) in the same column indicate significant differences (*p* < 0.05).

**Table 3 foods-10-02095-t003:** UPLC-MS identification of main phenolic compounds in Burdock root powder.

Peak	Rt (min) ^1^	Mass Parent ion *m*/*z*	Calc. MW	Tentative Compound
1	0.43	191	192	Quinic acid
2	0.53	191	192	Citric acid
3	2.175	163	164	*p*-Coumaric acid
4	2.836	353	354	3-caffeoylquinic acid (Chlorogenic acid)
5	3.131	353	354	4-caffeoylquinic acid (Cryptochlorogenic acid)
6	3.445	397	398	5-sinapoylquinic acid
7	4.104	515	516	3,4-di-*O*-caffeoylquinic acid
8	5.754	631	632	caffeoylquinic acid glycoside
9	6.112	615	616	1,5-di-*O*-caffeoyl-3-*O*-succinoylquinic acid
10	6.235	515	516	1,5-di-*O*-caffeoylquinic acid
11	6.486	367	368	*O*-Feruloylquinic acid
12	6.622	615	616	1,5-di-*O*-caffeoyl-4-*O*-succinoylquinic acid
13	6.826	301	302	Quercetin
14	7.266	631	632	3,4-di-*O*-caffeoyl-5-*O*-maloylquinic acid
15	7.687	515	516	3,5-di-*O*-caffeoylquinic acid
16	8.376	631	632	1,5-di-*O*-caffeoyl-4-*O*-maloylquinic acid
17	8.597	179	180	Caffeic acid

^1^ Rt is the retention time of UPLC-MS.

**Table 4 foods-10-02095-t004:** Elements in burdock root powder from Fengxian.

Element	Content (µg/g)
aluminum (Al)	192.48
barium (Ba)	2.90
beryllium (Be)	<0.50
calcium (Ca)	4205.58
cadmium (Cd)	<0.50
cobalt (Co)	<0.50
chromium (Cr)	20.45
copper (Cu)	13.23
iron (Fe)	58.49
potassium (K)	10,897.38
magnesium (Mg)	2424.32
manganese (Mn)	4.82
sodium (Na)	751.57
nickel (Ni)	<0.50
phosphorus (P)	3176.83
lead (Pb)	<0.50
antimony (Sb)	<0.50
titanium (Ti)	4.59
vanadium (V)	<0.50
zinc (Zn)	16.73

**Table 5 foods-10-02095-t005:** Volatile compounds of Fengxian burdock root powder.

Compounds	RT (min) ^1^	Area (%)
Alcohols		
hexyl alcohol	9.440	0.82
benzyl alcohol	14.863	0.66
phenethyl alcohol	17.957	1.00
1-penten-3-ol, 2-methyl-	20.939	0.41
elemol	27.986	0.33
1-hexadecanol	31.103	0.79
Aldehydes		
hexanal	7.828	4.64
furfural	8.664	0.42
(E)-2-hexenal	9.128	0.44
(E)-2-heptenal	12.034	0.56
benzaldehyde	12.340	1.11
(E,E)-2,4-heptadienal	13.963	0.56
phenylethanal	15.310	0.87
(E)-2-octenal	15.734	1.52
*n*-nonanal	17.486	2.38
(E)-2-nonenal	19.622	0.70
*n*-decanal	21.339	1.16
(E,E)-2,4-nonadienal	21.698	0.45
Acids		
hexanoic acid	12.828	3.05
2-ethyl-hexanoic acid	17.781	0.31
2-((2-chloroethoxy) carbonyl) benzoic acid	43.844	0.46
Alkenes		
3-(2-methylpropyl)-cyclohexene	13.328	3.53
1-dodecene	27.874	0.39
β-elemene	28.221	2.48
cadinene	28.668	0.40
α-cedrene	30.968	0.45
α-curcumene	31.062	1.01
(-)-alloaromadendrene	31.433	0.67
(Z,Z)-1,8,11-heptadecatriene	36.756	1.28
(Z,Z,Z)-1,8,11,14-heptadecatetraene	36.985	11.35
Esters		
sulfurous acid, hexyl octyl ester	25.039	0.40
oxalic acid, 2-ethylhexyl isohexyl ester	26.986	0.35
2,2,4-trimethyl-1,3-pentanediol diisobutyrate	34.585	1.05
Ketones		
octa-3-ene-2-one	14.987	1.03
(Z,E) and (E,E)-3,5-octadien-2-one	16.210	2.28
(E,E)-3,5-octadien-2-one	17.086	2.20
2,6-bis(1,1-dimethylethyl)-4-hydroxy-4-methyl-2,5-Cyclohexadien-1-one	30.615	1.18
Linear or aromatic hydrocarbons		
*n*-undecane	17.268	0.38
decamethyl-cyclopentasiloxane	19.239	2.07
naphthalene	20.851	0.76
dodecane	21.074	1.95
*n*-tridecane	24.715	1.89
1-methyl-naphthalene	24.892	0.43
5-(2-methylpropyl)-nonane	25.968	0.36
5-Ethyldecane	26.092	0.32
4,5-diethyl-octane	26.368	0.60
3,5-dimethyldodecane	27.156	1.31
2,6,10,14-tetramethyl-hexadecane	27.368	0.32
3-methyl-6-methylene-octane	27.674	0.43
tetradecane	28.145	2.98
nonyl-cyclopentane	29.827	0.56
2,6,10-trimethyltridecane	30.150	0.47
tetradecamethyl-cycloheptasiloxane	31.291	0.76
tetradecane	31.333	0.50
1,2,3,4,4a,5,6,8a-octahydro-4a,8-dimethyl-2-(1-methylethenyl)-naphthalene	31.691	0.62
n-tetradecane	33.503	0.49
hexadecane	34.427	0.47
Others		
dimethyl sulfide	4.440	1.28
3-methyl-5-hydroxy-isoxazole	10.169	0.32
2-sec-butyl-3-methoxypyrazine	20.157	0.31
2-methoxy-3-isobutyl pyrazine	20.475	0.33
2-(1,1-dimethylethyl)-6-methyl-phenol	26.915	0.39
2,4-di-tert-butylphenol	31.797	0.98
4-methyl-2,6-di-tert-butyl	31.991	2.19
2-(3,5-dimethyl-1h-pyrazol-1-yl) pyridine	36.150	0.92

^1^ RT (min): Retention time.

## Data Availability

Not applicable.

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
