# Peer review of "Comparison of Nutritional and Nutraceutical Properties of Burdock Roots Cultivated in Fengxian and Peixian of China"

_foods, 2021, doi:10.3390/foods10092095_

Round 1

Reviewer 1 Report

The reviewed manuscript concern comparative analysis of nutritional quality of different Burdock Roots. The work is interesting, but requires minor correction.

Please describe how volatile compounds from Burdock were isolated and collected. This information is not available in the manuscript or in the cited work.

Additionally, in the discussion on the composition of volatile compounds, the Authors mentioned dodecamethylcyclohexasiloxane, which is probably  supposed to come from the emission resulting from the degradation of the chromatographic column. This compound should be omitted from the analysis.

Author Response

  1. Please describe how volatile compounds from Burdock were isolated and collected. This information is not available in the manuscript or in the cited work.

Response:

Thank you for your professional suggestions. We also apologize for not mentioning the details of the separation and collection of volatile compounds from burdock. Firstly, we have added the details of the methodology used “A 5 g of sample was placed in an extraction bottle and was put in a water bath at 60°C for 30 min with magnetic stirring (300 rpm). The extraction needle was activated at the gas injection inlet for 20 min (250°C) before use” (Page 5, Lines 225-227). Secondly, we have added the heating conditions of GC-MS “We set the initial temperature at 50°C for 1 min and temperature increased at a rate of 5°C/min to 100°C and kept for 5 min. Afterwards, the temperature increased at a rate of 4°C/min to 140°C and kept for 10 min, further to 180°C for 10 min. Finally, it was increased at the same rate to 250°C and kept for 5 min” (Page 5, Lines 232-235).

  1. Additionally, in the discussion on the composition of volatile compounds, the Authors mentioned dodecamethylcyclohexasiloxane, which is probably supposed to come from the emission resulting from the degradation of the chromatographic column. This compound should be omitted from the analysis.

Response:

We apologized for adding the dodecamethyl cyclohexasiloxane compound which is likely from the emission resulting from the degradation of the chromatographic column. We have deleted the dodecamethyl Cyclohexasiloxane (Page 14, Table 5, Linear or aromatic hydrocarbons) and changed the proportion of “Linear or aromatic hydrocarbons” and “Others” in Figure 3. We have deleted the reference to dodecamethyl Cyclohexasiloxane and modified the proportion of Linear or aromatic hydrocarbons. “The linear/aromatic hydrocarbons and alkenes accounted for more than 48% of the volatile compounds, indicating that hydrocarbons are the main aroma components of BRP. The highest in proportion among linear or aromatic hydrocarbons was tetradecane, which accounted for 2.98% of the total” (Page 13, Lines 457-461).

Reviewer 2 Report

The manuscript “Comparative Analysis of Nutritional Quality of Burdock Roots Cultivated in Fengxian and Peixian of China” describes the differences between burdock roots from two Chinese regions. The manuscript is interesting but it requires numerous corrections.

Line 65 Replace 3,333.33 with 3,333. There’s no need to give that value so precisely.

Line 67 and 100 Country or county?

Lines 70-73 There’s no need to write the precise location of Fengxian and Peixian. Everyone can google these two regions.

Lines 86-97 No information about standards is given.

Lines 116-124 More details should be added.

Line 125 Replace “Amino Acid Analysis” with “Determination of Amino Acids”. The word “analysis” is used concerning samples and “determination“concerning analytes. So, you analyze burdock samples and determine amino acids.

Lines 127 and 133 Replace analysis with determination.

Line 128 Remove inorganic. The elements are not organic so there’s no need to write inorganic

Line 145 Delete “filters”.

Line 144 Which amino acids were determined using the DAD, which using the fluorescence detection?

Line 157 Write “for Identification of Phenolic Composition”.

Lines 162-164 Describe the gradient like in point 2.4. The present description is not clear.

Lines 165-166 The DAD was used in a range from 200 to 360 nm but a chromatogram at 280 nm is given (Figure 2). Why?

Line 168 Write 2.0 kV and 30 V.

Lines 158-172 Which operating mode was used in mass spectrometry?

Line177 Write 50 m x 0.25 mm x 0.25 µm. This is the usual way to write GC column dimensions.

Line 202 Write “The content of proteins was”

Line 203 Write “P-BRP”.

Line 216 Replace analysis with determination.

Line 221 Remove tryptophan. You mentioned it in line 220.

Line 224 Add a space before the word “determined”.

Lines 247-249 The difference between temperatures is only 1°C and this is the annual average. Can you conclude that it influenced the analyzed products?

Table 1 How many samples did you analyze? Only one? There are no error values. No statistical comparison of amino acid content is also given.

Line 267 Add a space after “enzymes” and “processes”.

Page 9 Line numbering should not be restarted on this page (which is important for the second review).

Lines 34-53 A comparison with the content of carotenoids should be discussed as it shows that there is no correlation with the antioxidant properties.

Lines 61-62 The text should be rewritten. For examples Figure 2 does not show LC-MS chromatogram but the text suggests it.

Line 69 The name of the second compound is different from that given in Table 3.

Line 74 Write “chlorogenic acid”.

Line 76 There is no peak 17 in the chromatogram.

Line 84 Remove inorganic. The elements are not organic.

Line 88 Add spaces after numerical values.

Lines 103-105 The sentence about chromium should be written after the sentence about calcium, magnesium, etc., but before the sentence about cadmium and lead. Moreover, you have no information about the oxidation state of Cr in your samples. Cr(VI) is carcinogenic. Take it into consideration.

Line 108 Replace “plumbum” with “lead”.

Line 110 Add a space after “elements”.

Table 4 Write cadmium with a small letter.

Table 4 Replace “plumbum” with “lead”.

Lines 117-142 I don’t see any sense in using the normalization procedure for the comparison of the content of such different compounds. Their ionization capabilities are different. The percentage area shows nothing.

Line 153 Remove a space before %.

Table 5 Reduce the accuracy of time (max. 3 decimal places) and area (max. 2 decimal places).

Table 5 Some names are written with capital letters, others with small letters. Be consistent.

Line 148 Write “were analyzed showing their nutritional quality differences.”.

Line 150 You discussed the influence of temperature and precipitation based on the annual values which are very similar concerning the temperatures. At the same time, no discussion is given concerning the type of soil.

Lines 161-162 There’s a statistical difference between the samples gathered in Fengxian and Peixian, but the second one still can be used in the industry. Perhaps the Peixian burdock is even better for the industry because the Fengxian burdock is better for direct consumption.

Lines 164-167 Who are W.Y.H. and X.X.Z.? I think that these authors have only one given name in lines 4-5. Reduce each second initial or add the second given name.

Line 165 Y.L. and W.Y.H. were responsible for validation but there’s no validation presented in the paper.

Reviewer 3 Report

The manuscript entitled "Comparative Analysis of Nutritional Quality of Burdock Roots Cultivated in Fengxian and Peixian of China" by Zhang et al., describes an interesting comparison regarding the chemical and nutritional quality of burdock roots from different provinces of China.

Considering the promising health-promoting properties of this plant-food, I think that the manuscript could be further considered after a round of major revision.

1) The title of the manuscript must be modified. Please, provide a clear take-home message to the reader (for example highlighting the nutraceutical profile).

2) In the Introduction section, the authors are invited to provide more information regarding food consumption and culinary uses of burdock.

3) Authors need to provide more information regarding sample collection. It is not clear where the authors sampled the plant material; is any voucher specimen available?

4) Authors should explain the choice of methanol 80% as extraction solvent. Probably, the use of water could be considered as a comparison, considering the eventual food consumption of this matrix.

5) Paragraph 2.3.: Authors must provide in detail the protocols for the determination of the different total compounds.

6) The antioxidant assays used by the authors are prone to interferences and have been banned by high-impact journals. Please, highlight these aspects in the manuscript.

7) Paragraph 2.6.: Authors should detail the level of confidence in annotation for their MS experiments. Have the authors used authentic standard compounds? The retention time has been compared with what?

8) No statistical analysis has been detected in Table 1.

9) Please, add "in vitro" together with the word antioxidants.

10) Authors should provide representative chromatograms in both LC-MS and volatile compounds analysis in order to check visual comparison between the different samples from different provinces. The lack of this work is the qualitative nature of the results, so authors should maximize the information provided to the readers in order to keep high the novelty.

Round 2

Reviewer 2 Report

The manuscript “Comparison of Nutritional and Nutraceutical Properties of Burdock Roots Cultivated in Fengxian and Peixian of China” has been greatly improved by the authors. There are only minor corrections to be done, i.e.:

  • The standards of amino acids are missing, add these standards in section 2.1.
  • There is no vitamin C standard mentioned in section 2.1 (although it is used as written in line 126). Give this information.
  • Water-soluble vitamin E is mentioned in section 2.1. Where was it used? In which procedure?
  • Lines 163-166 Rewrite that text, e.g. “The digested sample was cooled down before opening the tank cover to exhaust gases. The inner cover was rinsed with a small amount of water and the digestion tank was placed in an ultrasonic bath, degassed with ultrasounds for 2-5 minutes. The volume was made up to 50 mL with water and the sample was mixed. A blank test was performed at the same time using this procedure.”

Author Response

Please see the attachment,thanks!

Reviewer 3 Report

The authors have carefully revised the manuscript according to each major concern.

Author Response

Thank you for your previous suggestions, they are very important.
Thanks to your suggestions, I have found the deficiencies in my current work, which are of great guiding significance to my thesis writing and scientific research work.
Thank you again.